# What Does "ITS" Say about Hybridization in Lineages of *Sarsia* (Corynidae, Hydrozoa) from the White Sea?

Andrey Prudkovsky [1,*] , Alexandra Vetrova [2] and Stanislav Kremnyov [2,3]

1  Department of Invertebrate Zoology, Faculty of Biology, Lomonosov Moscow State University, Leninskie Gory 1/12, Moscow 119234, Russia
2  Department of Embryology, Faculty of Biology, Lomonosov Moscow State University, Leninskie Gory 1/12, Moscow 119234, Russia; lalavetrova@gmail.com (A.V.); s.kremnyov@gmail.com (S.K.)
3  Laboratory of Morphogenesis Evolution, Koltzov Institute of Developmental Biology RAS, Vavilova 26, Moscow 119334, Russia
*  Correspondence: aprudkovsky@wsbs-msu.ru

**Abstract:** Hydrozoans are widely known for their complex life cycles. The life cycle usually includes an asexual benthic polyp, which produces a sexual zooid (gonophore). Here, we performed an extensive analysis of 183 specimens of the hydrozoan genus *Sarsia* from the White Sea and identified four types of gonophores. We also compared the type of gonophore with haplotypes of the molecular markers COI and ITS. Analysis of COI sequences recovered that the studied specimens related to the species *S. tubulosa*, *S. princeps* and *S. lovenii*, and that the *S. lovenii* specimens divided into two COI haplogroups. More intraspecific genetic diversity was revealed in the analysis of the ITS sequences. The *Sarsia tubulosa* specimens divided into two ITS haplotypes, and presumably, hybrid forms between these lineages were found. For *S. lovenii*, we identified 14 ITS haplotypes as a result of allele separation. Intra-individual genetic polymorphism of the ITS region was most likely associated with intraspecific crossing between the different haplotypes. The diversity of the morphotypes was associated with the genetic diversity of the specimens. Thus, we demonstrated that the morphologically variable species *S. lovenii* is represented in the White Sea by a network of intensively hybridizing haplotypes. Hybridization affects the morphology and maturation period of gonophores and presumably affects the processes of speciation.

**Keywords:** hydroid; reduction of medusa; *Sarsia lovenii*; crossing; genetic polymorphism; gonophore

## 1. Introduction

Hybridization can be defined as reproduction between members of genetically distinct populations [1,2]. Hybridization may be the result of interactions involving a wide range of types and levels of genetic divergence between the parental forms [2]. In a broad sense, hybridization can occur between populations of the same species when there are constraints of free crossing and genetic divergence occurs. Hybridization has been considered as one of the mechanisms that influence the process of speciation [2,3]. The hybrid can reproduce either with its parental lineages (backcrossing or introgression) or only with similar hybrids [3]. In both cases, hybridization can lead to the emergence of novel features as well as new species altogether. Reticulate evolution caused by hybridization has played an important role in the diversification of several anthozoan genera [4–7]. Much less is known about the importance of hybridization in the Medusozoa taxa, including Hydrozoa [8–13] and Scyphozoa [14].

The complex life cycle of Hydrozoa includes pelagic medusa and sessile polyp stages [15–18]. Free-swimming medusae detach from benthic polyps or colonies, grow and spawn gametes after maturation. A ciliated larva, i.e., a planula, develops from the fertilized egg, settles and undergoes metamorphosis into the new polyp. However, reduction in the medusa stage is a widespread evolutionary trend among Hydroidolina and occurs

independently in many phylogenetic lineages [19–21]. Reduced gonophores lose many of the features of the medusa and usually produce gametes whilst staying attached to the parental colony.

Recently, an unusual morphogenetic polymorphism was found in the hydroid *Sarsia lovenii* (M. Sars, 1846) (Corynidae) [13]. According to traditional views, colonies of *S. lovenii* produce reduced medusae named medusoids [22]. Medusoids form "gonads" without breaking away from the parental colony and lack ocelli and tentacles. Recently, it was demonstrated that *S. lovenii* has two morphotypes of gonophores; some colonies produce free-swimming medusae, while others produce medusoids [13]. The studied morphotypes belong to different genetic haplogroups, but the genetic distances between these haplogroups are minimal and correspond to the level of intraspecific variability. The possibility of crossing between these haplogroups has also been experimentally proven. The results obtained were interpreted as a case of incipient speciation [13]. However, little attention was paid to the crossing (intraspecific hybridization) of different lineages of *S. lovenii* in the sea.

The aim of our work is a detailed analysis of the morphogenetic diversity of hydrozoans *Sarsia* Lesson, 1843, in the White Sea, including a search for the natural crossing (intraspecific hybridization) of lineages of *S. lovenii* using a region of internal transcribed spacers of the ribosomal operon (ITS1-5.8S-ITS2).

## 2. Materials and Methods

### 2.1. Sampling and Experimental Cultures

The material was collected near the Pertsov White Sea Biological Station (Lomonosov Moscow State University) (66°34′ N, 33°08′ E) in 2015–2021 (Figure 1, Table S1).

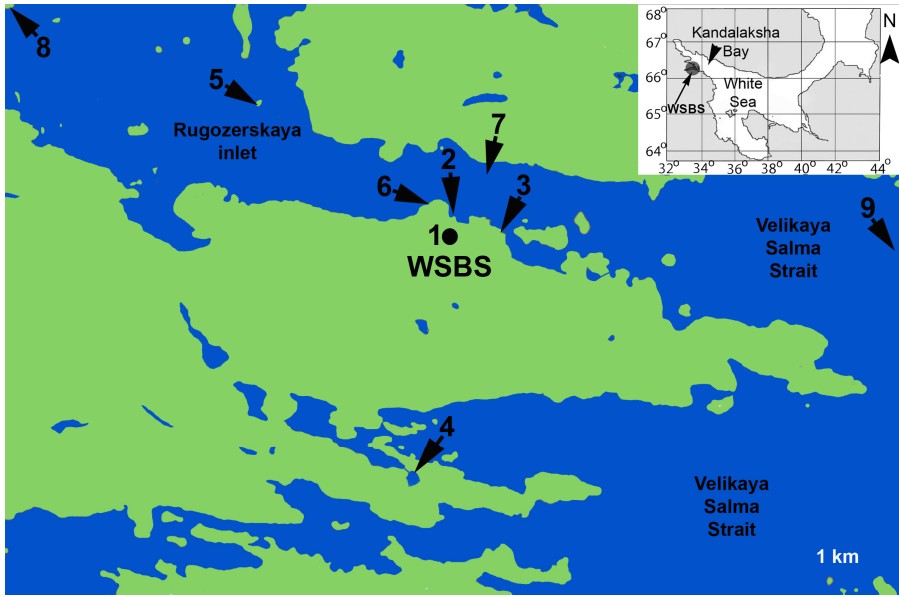

**Figure 1.** Sampling localities in the White Sea. Location of the WSBS in the White Sea is shown at the inset. Abbreviations: WSBS—White Sea Biological Station; 1—aquarium at WSBS; 2—pier of WSBS; 3—Eremeevskie rapids; 4—saline lake at the Green Cape; 5—location "Luda"; 6—location "Krest"; 7—Rugozerskaya inlet, depth 5–15 m; 8—Polovye islands; 9—Velikaya Salma Strait, depth 40–60 m.

Colonies were collected on different substrates near the shore or captured by trawling or diving (Table S1). Medusae were collected manually near the surface of the water. Two hydroids were collected outside the White Sea: a colony of *Sarsia lovenii* with medusoids was collected in the Barents Sea (Dalnezelenetskaya inlet), and a medusa of *S. lovenii* was collected in the Bering Sea (Senyavin Strait). Medusae and colonies were photographed alive and fixed in ethanol (96%) for molecular phylogenetic analysis. Additionally, the

collected specimens were used for experimental cultures or for crossing experiments. For experimental crossing, ready-to-spawn medusae and medusoids of *Sarsia* spp. were collected and maintained in small containers (200 mL) with filtered (0.2 microns) seawater at a temperature of 10–12 °C. Females of *S. lovenii* with a "medusoid" morphotype were placed together with male medusae of *S. lovenii*, and female medusae of *S. lovenii* were placed with male medusoids of *S. lovenii* (Table 1). In addition, a female medusa of *Sarsia tubulosa* (M. Sars, 1835) was crossed with a male medusoid of *S. lovenii*. About seventy experimental cultures were kept in the laboratory for several months or up to 1.5 years (Table S1). Polyps were fed with *Artemia nauplii*. To encourage the production of gonophores, the temperature in the aquarium was reduced to 1–3 °C for about two months and then gradually increased at a rate of 1 °C per week. In addition, changing about half of the water in the aquarium stimulated the hydroids to produce gonophores. For 32 colonies, we traced the development of gonophores. We photographed the growing gonophores weekly for 1–2 months from the beginning of their production either to the detachment of medusae (4 colonies) or to the appearance of gonads on the manubrium of the gonophores (28 colonies) (Table S1). After that, we continued to track the gonophores with gonads for a month and in some cases observed eggs and mature sperm in the gonads. Two experimental cultures, F1, with mature sperm were used for backcrossing with female medusa of *S. lovenii*. The descendants of this cross were cultivated for several weeks and then were fixed in ethanol (96%).

**Table 1.** Schemes of crossing experiments and related DNA isolates.

| Schemes of Crossing | DNA Isolates |
|---|---|
| *S. lovenii* medusa female × *S. lovenii* medusoid male | H140, H144, H151, H153, H157, H168 |
| *S. lovenii* medusa male × *S. lovenii* medusoid female | H149, H150, H158, H159, H170 |
| *S. lovenii* medusa female × *S. lovenii* male | H122 |
| *S. lovenii* medusoid female × *S. tubulosa* male | H163 |
| *S. tubulosa* female × *S. tubulosa* male | H249 |
| *S. lovenii* medusa female × *S. lovenii* medusa male | H238, H251 |
| *S. lovenii* hybrid F2: hybrid F1 males (H159 + H140) × *S. lovenii* medusa female | H233, H236 |

In total, 183 specimens were used for analysis (Table 2); 143 specimens were collected in the sea (including 42 medusa specimens and 101 polyp specimens) and 40 specimens were sampled in an aquarium (including 18 specimens obtained by experimental crossing). For 99 specimens, we observed mature gonophores (including medusa specimens) or experimentally induced their formation.

**Table 2.** Number of specimens of *Sarsia* spp. collected in different years.

| Year of Collection | Sampling Location | Number of Specimens | Number of Medusa Specimens/Number of Polyp Specimens with Mature Gonophores |
|---|---|---|---|
| 2015 | Aquarium | 6 | 0/1 [13] |
| 2016 | In the sea, WSBS | 9 | 4/0 [13] |
| 2017 | In the sea, WSBS | 23 | 22/1 [13] |
| 2018 | In the sea, WSBS | 5 | 1/4 [13] |
| 2019 | In the sea, WSBS, Bering Sea, Barents Sea | 43 | 1/10 |
| 2019 | Aquarium | 17 | 0/5 |
| 2019 | Aquarium (crossing experiment) | 13 | 0/11 |
| 2020 | In the sea, WSBS | 43 | 14/8 |
| 2020 | Aquarium (crossing experiment) | 4 | 0/0 |
| 2021 | In the sea, WSBS | 20 | 0/17 |

## 2.2. Morphological Analysis

We observed the process of gonophore development from the moment of their appearance to the period of gonad formation and spawning or until the moment of detachment of the gonophore from the parent colony. Based on the results of this observation, we selected the features for the morphotype delimitation. To distinguish the different morphotypes of gonophores in the collected specimens and the experimental colonies we analyzed, we asked the following questions: does the gonophore detach from the parental polyp as a free-swimming medusa or not; are the edges of the bell with tentacles bent inward or not; does the bell unfold before detachment or not; is there the presence of tentacles and ocelli on the tentacular bulbs; is there the presence of an incomplete nematocyst ring in epidermal part of bulbs; what is the shape of the bell and the size of the manubrium; is there the presence or absence of gonad at the gonophore before the detachment.

## 2.3. Molecular Analysis

DNA was isolated with a Diatom kit (Diatom DNA Prep 100 kit, Isogen Laboratory, Moscow, Russia) according to the manufacturer's protocol. The cytochrome c oxidase (COI) subunit fragment I and internal transcribed spacers of the ribosomal operon 18S-ITS1-5.8S-ITS2-28S rRNA (ITS) were amplified from isolated DNA with the following primers pairs: SR6R (AAGWAAAAGTCGTAACAAGG) and LR1 (GGTTGGTTTCTTTTCCT) for 18S-ITS1-5.8S-ITS2-28S rRNA [23,24] with a program of 95 °C for 5 min followed by 34 cycles of 15 s at 94 °C, 30 s at 52 °C and 60 s at 72 °C and then a final extension of 5 min at 72 °C; and SAR-F (TTTGGGGCTTTCGCCGGTAT) and SAR-R (CAGGATCACCTCCTCCTGC) for COI (*Sarsia*-specific primers, current study) with a program of 95 °C for 5 min followed by 34 cycles of 15 s at 95 °C, 30 s at 50 °C and 60 s at 72 °C and then a final extension of 5 min at 72 °C. The polymerase chain reaction was carried out in a reaction volume of 20 μL, which included 4 μL of 5× Screen Mix solution (Eurogen, Moscow, Russia), 0.5 μL of each primer, 1 μL of DNA and 14 μL of sterile water. Amplification was also carried in a volume of 25 mL, which included 5 mL of 5× Taq Red Buffer (Evrogen Lab, Moscow, Russia), 0.5 mL of polymerase (HS-Taq Polymerase by Eurogen Lab), 0.5 mL of dNTP (50 μM stock), 0.3 mL of each primer (10 μM stock), 1 mL of DNA and 17.7 mL of sterile water (MilliQ). Sequencing was carried out at Evrogen (Moscow, Russia) in an ABI Prism 3500 Genetic Analyzer (Applied Biosystems, Waltham, MA, USA). New COI and ITS sequences for 140 specimens were obtained (Table S1). Previously obtained sequences were also used for the analysis (the list of specimens can be seen in [13]). Accession numbers of the sequences generated in the present study are listed in Table S1, i.e., accession numbers COI (from OQ859724 to OQ859863) and accession numbers ITS (from OQ862838 to OQ863014).

## 2.4. DNA Cloning

Specimen H122 was an experimental hybrid of a female medusa *S. lovenii* and a male medusa *S. lovenii*. Specimen H144 was an experimental hybrid of a female medusa *S. lovenii* and a male medusoid *S. lovenii*. The ITS of the H122 and H144 specimens were isolated from the genome DNA samples with a gene-specific SR6R and LR1 primer pair. Amplified fragments were cloned into the pAL-TA vector (Evrogen, Moscow, Russia). Three clones were sequenced from each plate.

## 2.5. Phylogenetic Analysis

Sequences were assembled and checked for improper base-calling with the Codon-Code Aligner software V. 6.0.2 (www.codoncode.com/aligner (accessed on 13 April 2013)). Sequences were aligned using the MUSCLE [25] algorithm in the MEGA 6 software [26]. The final alignments resulted in a dataset comprising 624 bp for the COI. JModelTest 2 [27] was used to estimate the best substitution model for each partition based on the Bayesian information criterion (BIC). The GTR + G model was found to be optimal for the COI dataset. Bayesian phylogenetic trees were built in PhyloBayes 3.3 [28]. The analysis was performed with random starting trees and 10 million generations. Two MCMC chains

were run in parallel, and the analyses were stopped when the maximum discrepancy of bipartitions between chains was below 0.01. The final phylogenetic tree images were rendered in FigTree 1.4.0. Maximum-likelihood phylogenetic analysis was performed in the IQTree v.2.0-rc2 software [29] with the standard algorithm. The best model of nucleotide substitution (GTR + F + G4) was chosen using ModelFinder [30] according to the Bayesian information criterion (BIC). One thousand bootstrap replicates were generated for the analysis.

A haplotype network for the COI dataset was constructed using the TCS network inference method [31] within the PopART software 1.7 (http://popart.otago.ac.nz/index.shtml (accessed on 19 December 2017)). According to a constraint of the method, we used a reduced COI dataset without undefined states of nucleotides. Haplogroups of *S. lovenii* were identified in accordance with [13] or according to the morphology of the specimens.

For the analysis of ITS sequences with heterozygotes, we used Champuru v. 1.1 [32] (https://eeg-ebe.github.io/Champuru/input.html (accessed on 15 February 2023)), a computer software program for unraveling mixtures of two DNA sequences of unequal lengths. Champuru makes it possible to determine the haplotypes of heterozygous individuals without cloning simply by analyzing the patterns of double peaks in the forward and reverse chromatograms (marked in our results as phase-1 and phase-2). This method is well suited for unraveling mixtures of haplotypes that differ only in one deletion locus, which is typical for different interlineage hybrids of *S. lovenii*. Sequences with one heterozygote were manually divided into two alleles (marked in the results as allel-1 and allel-2). We trimmed the ends with unknown bases to align the length of all the sequences. In addition, we excluded several short sequences of less than 500 bp (H150, H157, H161, H170) and sequences with two single heterozygotes (H97, H118, H153, H184) from the analysis. ITS fragments of several of the *S. tubulosa* specimens, namely those containing many single heterozygotes, also could not be divided to haplotypes. The final alignments resulted in a dataset comprising 509 bp for the 214 sequences/152 specimens. We exported the dataset in RDF format (Roehl data file) using the DnaSP 5.10 software with the option «considered sites with gaps/missing» [33]. A haplotype network was constructed using the median joining algorithm [34] within the NETWORK 10.2.0.0 software (Fluxus Technology Ltd., Sunnyvale, CA, USA, www.fluxus-engineering.com (accessed on 17 February 2023)).

## 3. Results

### 3.1. Morphotypes of Gonophores in Specimens of Sarsia spp.

We identified the following four morphotypes of gonophores: free-swimming medusa (53 specimens), medusoid (17 specimens), attached medusa (17 specimens) and "abnormal medusoid" (12 specimens) (Figures 1 and S1, Table S1 [13]). Three species of *Sarsia* (*S. lovenii*, *S. tubulosa* and *S. princeps*) in the White Sea produced free-swimming medusae. The remaining morphotypes belonged to *S. lovenii*.

Morphotype I, medusoid, was characterized by the absence of tentacles and eyes on tentacular bulbs (Figure 2A). The edges of the bell were not wrapped inside the subumbrellar cavity but were located freely at the developing gonophore. Tentacular bulbs were present but were significantly reduced compared to the other morphotypes and lacked a c-shaped nematocyst zone. The bell was oval when observed from the side. It was narrowed in the proximal (apical) part, where the bell was attached to the parental polyp, and in the distal, where the tentacular bulbs were located. The gonad encircled the manubrium. The manubrium with gonad occupied a large part of the subumbrellar cavity. The gonad on the manubrium appeared early in the development, occasionally covering even the most distal area of the manubrium. There was no functional mouth; the medusoid did not feed. Mature gametes fell into the bell cavity. When the medusoid was ripe, it was possible to observe a series of bell contractions, due to which gametes were expelled from the subumbrellar cavity. The colonies of *S. lovenii* with ripe medusoids were collected in the sea from June to July, and the medusoids were also produced by some experimental colonies (Figure S1, Table S1).

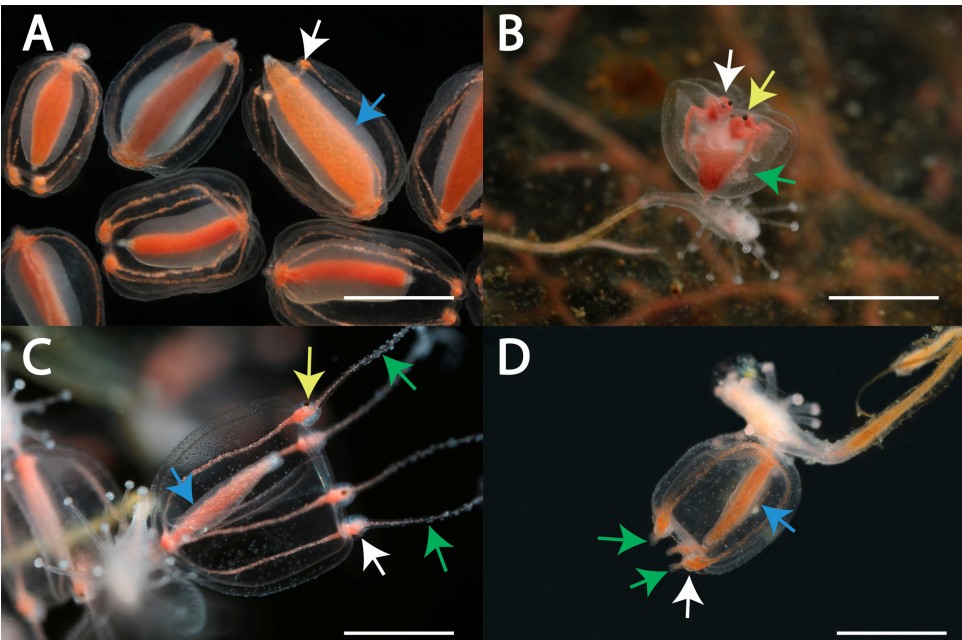

**Figure 2.** Four morphotypes of gonophores in specimens of *Sarsia lovenii* from the White Sea: morphotype I—medusoid specimens H347 detached from the polyps (**A**), morphotype II—bud of free-swimming medusa specimen H167 (**B**), morphotype III—attached medusa specimen H140 (**C**), morphotype IV—"abnormal medusoid" specimen H341 (**D**). Abbreviations: yellow arrows indicate ocelli, green arrows indicate tentacles, white arrows indicate tentacular bulbs and blue arrows indicate gonad covering manubrium. Scale bars 1 mm.

Morphotype II, medusa, detached from the parental colony when its main parts were shaped, such as the bell, a manubrium and tentacular bulbs with ocelli and tentacles. Further growth and maturation of the gonad occurred in the free-swimming feeding medusa. The medusae of *Sarsia* spp. could be identified by morphological characteristics such as the size of the bell, the morphology of the tentacular bulbs and apical knob and the position of the gonad over the manubrium [13,22]. We observed the development of medusa buds in *S. lovenii*. Late medusa buds were characterized by tentacles and ocelli on the tentacular bulbs, the edges of the bell were bent inward, and the tentacles were inside the bell (Figure 2B). The medusa turned the tentacles out shortly before the detachment from the parental polyp, when the bell began to contract. The medusa broke away from the colony within a day after the bell started to contract and the bell edges started to unfold. Newborn medusae were able to feed immediately after separation from the colony. Medusa buds of *S. lovenii* were collected at the sea in March and early April and were also obtained on some experimental colonies (Figure S1, Table S1). Since April, *S. lovenii* medusae were found in the water column, and they spawned in June.

Morphotype III, attached medusa, was similar to a new-born free-swimming medusa in terms of the shape of the bell (Figure 2C). Such attached medusae had relatively short tentacles and tentacular bulbs with ocelli. Late gonophores had a fully expanded bell, outward-located tentacles and were capable of periodic contraction. A mouth opening was present at the end of the manubrium. The gonad covered the proximal and middle parts of the manubrium as a tube. The gonophores remained attached to the parental colony for a long time despite the presence of tentacles, tentacular bulbs with ocelli and gonad. However, if the parental polyp was resorbed, the gonophore could potentially break away from the parental colony and swim near the bottom and even be able to feed. The size of such medusae did not exceed 2–3 mm. In the experiment, the medusae detached mainly after mechanical manipulations with the colony during observation. Some gonophores with ocelli, spread tentacles and a gonad remained on the parent colony for up to a month until the maturation of gametes in the gonad. The morphotype attached medusa was

formed on colonies that were obtained as a result of the experimental crossing between the medusa morphotype and the medusoid morphotype. The colonies were obtained by the intraspecific crossing of individuals of *S. lovenii* and by interspecific crossing between the medusoid of *S. lovenii* and the medusa of *S. tubulosa*. Moreover, gonophores with such a morphotype were found in the sea on 12 May 2021 (Figure S1, Table S1).

Morphotype IV, abnormal medusoid, differed from the typical medusoid by well-developed tentacular bulbs that often looked like short rod-shaped tentacles (Figure 2D). A c-shaped zone of nematocysts was also visible in the bulbs. The shape of the bell was the most similar to the morphotype attached medusa. However, such gonophores lacked ocelli on the tentacular bulbs. Abnormal medusoids could break away from the parental colony because of a mechanical impact and move near the bottom of the experimental bowl. Abnormal medusoids were found on colonies of *S. lovenii* in the sea on 12 May 2021 (Figure S1, Table S1).

### 3.2. Analysis of COI

The analysis of the molecular phylogenetic tree and the haplotype net of the mitochondrial marker COI allowed us to divide the collected specimens into three species: *S. tubulosa*, *S. lovenii* and *Sarsia princeps* (Haeckel, 1879) (Figure 3, Table S2). Moreover, the specimens of *S. lovenii* formed two haplogroups: haplogroup COI-1 and haplogroup COI-2 (Figure 3B; see also [13]). Haplogroup COI-1 included specimens of morphotypes I, III and IV (medusoid, attached medusa and abnormal medusoid). Haplogroup COI-2 included specimens with morphotypes II and III (medusa, attached medusa).

The diversity of the COI haplotypes of *S. lovenii* in the White Sea was low, and most of the specimens belonged to the two widespread haplotypes. Some haplotypes of *S. lovenii* that had unique substitutions were mainly from other locations, such as the Barents Sea (OQ859859), North Sea (KT981910) and Canada, Nunavut (MG422634). Specimen H248 (OQ859798), found in the deep-water part of the White Sea, also had a haplotype that was dissimilar from the littoral specimens but was identical with the medusa specimen from Canada. We assigned it to the haplogroup COI-2 because the specimen from Canada was a medusa. There were also some unique haplotypes of *S. lovenii* collected in the shallow part of the White Sea, which were adjacent to one or another haplogroup. The medusa specimen from the Bering Sea (H97) had unique haplotype (OQ859863) that was dissimilar from the other medusa haplotypes *S. lovenii*. Being closer to haplogroup COI-1, it nevertheless had the morphotype II. The specimens of *S. tubulosa* from the White Sea were grouped with some haplotypes from the North Sea (Figure 3B). The specimen of *S. princeps* from the White Sea was grouped with some haplotypes from the North Atlantic, including Canada's water and the Iceland Sea (Figure 3B).

### 3.3. Analysis of ITS

For the analysis, a dataset with a length of 509 nucleotides was built. Within the dataset, 31 positions were variable (Figure 4, Table S3). We distinguished the groups of specimens belonging to certain haplotypes as well as specimens with heterozygous states of certain loci, which may indicate the processes of crossing between individuals with different haplotypes in populations.

The heterozygous specimens of *Sarsia tubulosa* had ITS sequences with several single heterozygous loci. Such sequences could be the result of a combination of potential alleles, and the number of combinations increased rapidly with the increase in the number of heterozygotes. Therefore, we only used the specimens of the two main haplotypes without heterozygous loci (st-1 and st-2) in the network of haplotypes (Figure 4, Table S4). The differences between them were in seven loci. The heterozygotes from these loci found in 10 specimens probably indicated hybridization between the lineages st-1 and st-2 (Figure 4, Table S4). Two haplotypes of *Sarsia tubulosa* and heterozygous specimens had gonophores of morphotype II (free-swimming medusa). We did not find any significant differences in the morphology of the medusae of different haplotypes (Figure S1, Tables S1 and S4).

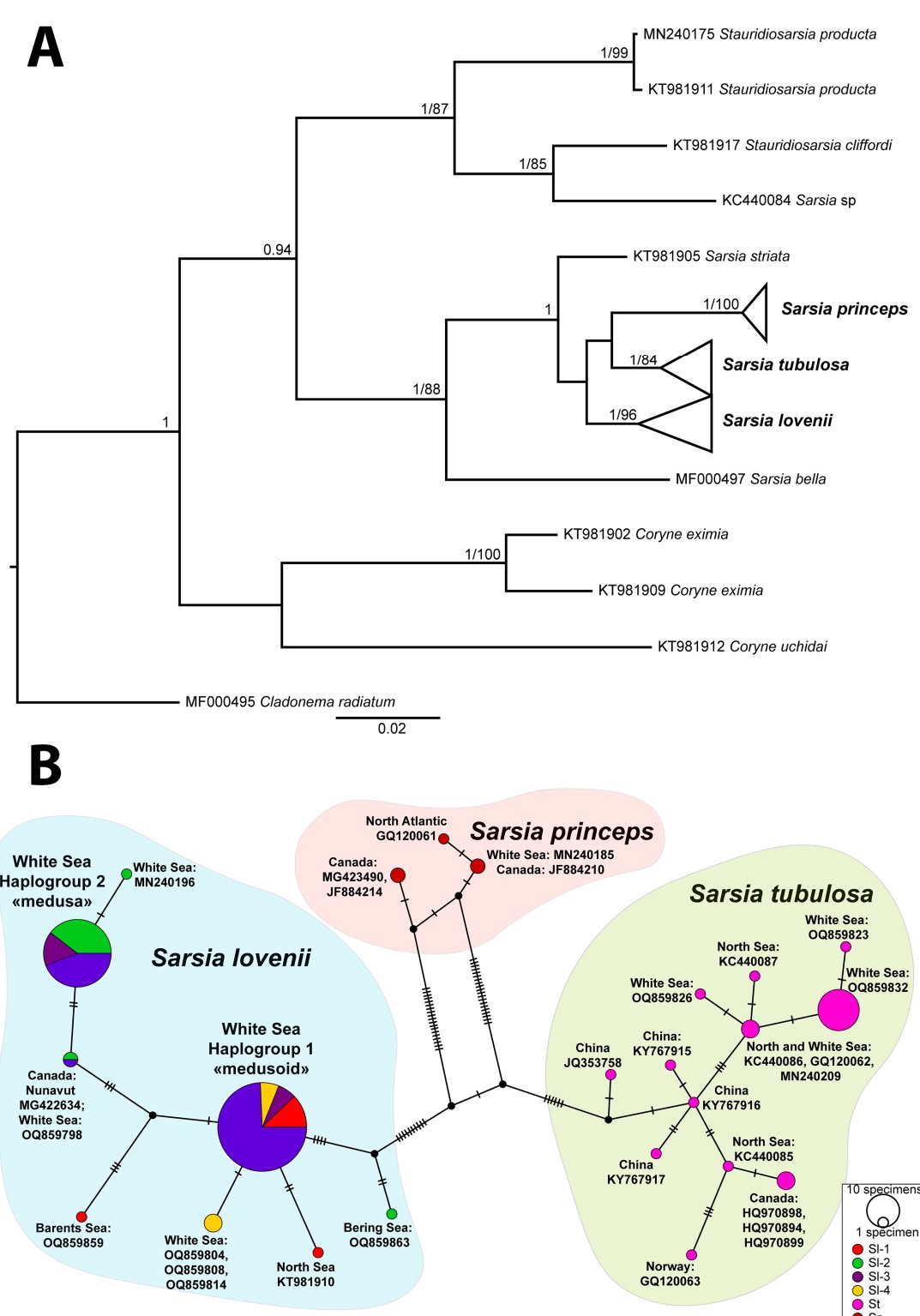

**Figure 3.** Phylogenetic analyses of COI dataset: (**A**) Bayesian and maximum-likelihood phylogenetic hypotheses. Numbers near branches show posterior probabilities (>0.95) and bootstrap values (>70). (**B**) Haplotype network. Abbreviations: morphotypes of gonophores *Sarsia lovenii* (Sl-1–Sl-4) relate to morphotypes I–IV in the Figure 2. St—*Sarsia tubulosa*; Sp—*Sarsia princeps*; Unknown—specimens lacked ripe gonophores.

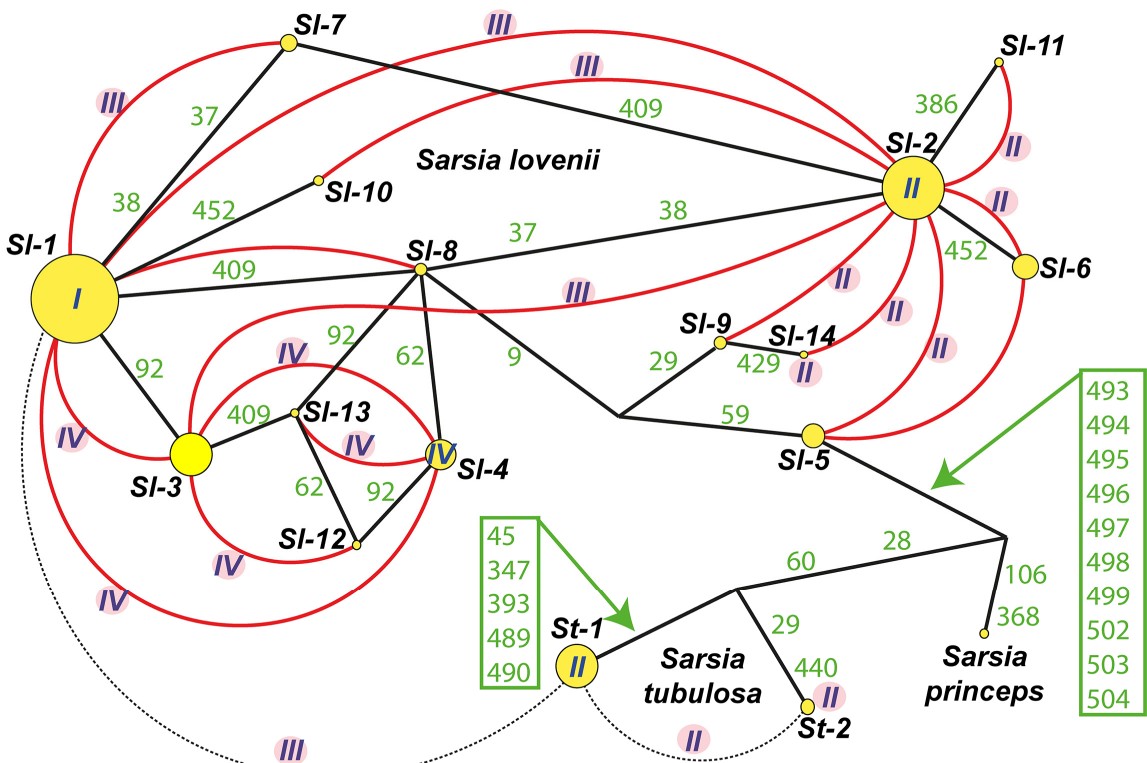

**Figure 4.** Haplotypes net for ITS dataset including *Sarsia lovenii*, *Sarsia tubulosa* and *Sarsia princeps*. List of specimens see in Table S4. Abbreviation: Sl-1–Sl-14—haplotypes *S. lovenii*, St-1–St-2—haplotypes *S. tubulosa*; I–IV—morphotypes of gonophores for haplotypes or hybrid specimens (I—medusoid, II—medusa, III—attached medusa and IV—abnormal medusoid); Green numbers—variable loci in ITS dataset (see Table S3); Yellow circles—haplotypes ITS, size of circle relates to number of specimens; Black solid lines—connection between haplotypes; Red curves illustrate crossing between haplotypes; The dotted line indicates crossing between st-1 and st-2 haplotypes, as well as experimental interspecies hybrid between St-1 and Sl-1.

A large number of heterozygous loci were obtained for an experimental interspecies hybrid between *S. tubulosa* and *S. lovenii* (Figure 4: hybrid Sl-1/St-1). Due to the presence of several deletion zones, we could not determine the state of some loci when separating the alleles using Champuru v. 1.1 [32]. Such hybrid colonies produced an attached medusa (morphotype III) (DNA isolate H163: Figure S1, Table S1).

Several heterozygous specimens of *Sarsia lovenii* had ITS sequences with a group of double peaks in the chromatograms. We managed to unravel the mixture of two alleles for specimens with heterozygotes using Champuru v. 1.1 [32] as well as by the cloning of two DNA isolates. As a result, we identified 14 haplotypes for *Sarsia lovenii* (Figure 4, Table S4). Only three haplotypes (Sl-1, Sl-2 and Sl-4) included specimens with ITS sequences without heterozygotes. The remaining specimens had ITS sequences with heterozygotes and became part of different haplotypes, these being divided into alleles. Hydroids with the Sl-1 haplotype produced normal medusoids (morphotype I). The Sl-2 haplotype was found in the free-swimming medusae or in the hydroids that produced free-swimming medusae. In addition, free-swimming medusae were registered in the specimens with a mix of the allele Sl-2 and any of the alleles Sl-6, Sl-11, Sl-5, Sl-9 or Sl-14.

A hybrid specimen was obtained in an experimental crossing between a medusa with the haplotype Sl-2 and a medusa with the haplotype Sl-14 (DNA isolate H122). The resulted hybrid colony produced free-swimming medusae (morphotype II). The experimental crossing of specimens with the haplotypes Sl-1 and Sl-2 resulted in hybrid colonies that produced attached medusae (morphotype III) (Table 1). Colonies of hydroids with such a morphotype of gonophores were also found in the sea in May (DNA isolates H335, H337,

H338). In addition, attached medusae were registered in the specimens with a mix of the allele Sl-2 and allele Sl-10 or in mix of allele Sl-1 and allele Sl-7 (Figure 4, Table S4).

Colonies of hydroids with the haplotype Sl-4 produced medusoids of an abnormal structure (morphotype IV). Several more heterozygous specimens for ITS had the same morphotype. Being separated into alleles, these sequences became part of the haplotypes Sl-1, Sl-3, Sl-4, Sl-12 and Sl-13 (Figure 4, Table S4).

## 4. Discussion

### 4.1. Morphotypes of Gonophores in Sarsia lovenii in the White Sea and Period of Reproduction

In this work, we found four morphotypes of gonophores in *Sarsia lovenii* (medusoid, free-swimming medusa, attached medusa and abnormal medusoid) compared to the three that were described earlier [13]. Early gonophores of different morphotypes are similar but can be distinguished when the main morphological characteristics appear, such as tentacular bulbs, tentacles, ocelli and gonads. Some features may vary. Mature medusoids (morphotype I) vary in size, color, the shape of the bell and the size of the manubrium and gonad. The shape of the bell in medusoids varies from almost spherical (H139) to elongated (H347) (Table S1). However, they never have tentacles on the bulbs. An important feature of free-swimming medusa (morphotype II) is that they break away from the parental colony long before the appearance of the gonad and grow in the water column. Medusae of morphotype III can break away from the parental colony too; they can swim near the bottom and even feed. However, attached medusae become free only as a result of mechanical action after the appearance of the gonad. Most of the medusae in this experiment broke away from the parental colony when the experimental hydroids were pulled out of the aquarium for observation. The size of such medusae (morphotype III) is much smaller (1–3 mm) than the size of mature free-swimming medusae of morphotype II (7–16 mm) [13]. Morphotype IV, which we called "abnormal medusoid", differed from typical medusoids due to the presence of elongated rod-like tentacular bulbs and a nematocyst zone at the tentacular bulbs (Figure 2D). On the other hand, the abnormal medusoids did not have ocelli at the tentacular bulbs and thus differed from attached medusae. This morphotype has not been previously described for *Sarsia* hydroids and, thus, it is not yet known whether it occurs outside the White Sea.

In addition to the morphological differences, the different morphotypes of gonophores had different periods of occurrence in the sea (present data [13]). Gonophores with morphotype II were found on the hydroids in March and early April. Since April, *S. lovenii* medusae were present in the water column, and spawning occurred in June [13]. Gonophores with morphotypes III (attached medusae) and IV (abnormal medusoids) were found on the hydroids only in the first half of May. Mature gonophores with morphotype I (medusoids) were found on the hydroids in June and July. Differences in the reproduction period for different morphotypes of *S. lovenii* can lead to partial genetic isolation and maintain the morphogenetic diversity of the species.

### 4.2. Crossing Experiments

Here, we experimentally confirmed the possibility of crossing between different haplotypes of *S. lovenii* and of interspecific hybridization between *S. lovenii* and *S. tubulosa* (Table 1, Figure S1). The possibility of hybridization between medusae and medusoids of *S. lovenii* has already been proven [13], but in this study, we confirmed the results in several repetitions. The hydroids obtained by crossing between free-swimming medusae and medusoids produced gonophores with the morphotype attached medusa.

We also performed backcrossing between a first-generation hybrid (F1) and the medusa *S. lovenii*. The crossing was successful. However, the poor survival of the resulting hybrids F2 did not allow a quantitative analysis of the different alleles in the descendants. Nevertheless, our results confirmed the possibility of such crossings in the sea.

*4.3. Phylogeny, Species and COI Haplogroups of Sarsia spp. in the White Sea*

According to the COI analysis, the collected specimens were attributed to three species: *S. lovenii*, *S. tubulosa* and *S. princeps* (Figure 3A). Previously, a detailed multigenic phylogenetic analysis of the species composition of the *Sarsia* genus was performed [13]. In addition, the COI dataset was used for the species delimitation analysis as well as the intraspecific diversity analysis of *S. lovenii*. Here, we did not conduct a detailed analysis of these results since the new data corresponded to previously obtained data. The *Sarsia lovenii* specimens were attributed to two main haplogroups [13]. Nevertheless, the support of two *S. lovenii* clades on the multigenic phylogenetic tree was low [13] due to the small genetic distances between the lineages and due to the presence of specimens with heterozygotes. The delimitation of two haplogroups of *S. lovenii* was supported by the analysis of morphotypes. The specimens with medusoids were assigned to haplogroup COI-1, and those with free-swimming medusae were assigned to haplogroup COI-2 [13]. However, here, we demonstrated that each haplogroup included specimens with several types of gonophores (Figure 3B). Specimens with an attached medusa (morphotype III) were present in each haplogroup. Attached medusae were produced in the experiment by hybrids between medusae and medusoids of *S. lovenii*. Given that mitochondrial genes are inherited on the maternal side, we believe that hybridization between the haplogroups went in both directions. In addition, the haplogroup COI-1 also included specimens with the morphotype of gonophore IV. A special position was occupied by a specimen from the Bering Sea, which had a medusa as a gonophore but was more closely related to haplogroup COI-1. Perhaps the medusa in the evolution of the species *S. lovenii* could be reduced to a medusoid and then recover again. However, to understand the phylogeography of the species *S. lovenii*, more specimens from different locations are required.

*4.4. Hybridization or Intragenomic Polymorphism?*

Nuclear ribosomal DNA (nrDNA) is the genomic region in which the RNA components of ribosomes are encoded [35–39]. Eukaryotic nrDNA comprises a multigene family including transcribable rRNA genes (18S rRNA, 28S rRNA and 5.8S rRNA) separated by internal transcribed spacers (ITS1 and ITS2) and an intergenic spacer (IGS) that are located downstream of the 18S rRNA gene and upstream of 28S rRNA gene. These genes cluster in large tandems located on certain chromosomes to form nucleolus-organizing regions. Ribosomal genes and their associated spacers are arranged into one or more large arrays consisting of hundreds or thousands of tandemly repeated copies. During evolution, coding regions (18S and 28S rRNA) have remained more conserved than non-coding regions (ITS and IGS). There is a considerable precedent for the use of ITS sequence divergence to infer relationships at or below the species level in a wide variety of taxonomic groups, most notably in plants and fungi [40–43]. Sometimes ITS sequences are used in recovering the phylogeny of cnidarian taxa such as corals [44] and hydrozoans [45–50]. In addition, the ITS region is used to study intraspecific genetic heterogeneity [51]. In our study, we presented the results of a detailed analysis of the ITS marker in 183 specimens of *Sarsia* from the White Sea. We found three pure haplotypes of *S. lovenii* (Sl-1, Sl-2 and Sl-4), two haplotypes of *S. tubulosa* (St-1 and St-2) and one haplotype of *S. princeps*. All these haplotypes did not contain heterozygotes. We also found intra-individual polymorphism in the structure of the ITS for *S. lovenii* and *S. tubulosa*.

The ITS region can be hypervariable and prone to insertions and deletions, which can result in alignment ambiguities [44,51]. When analyzing the chromatograms, we encountered ambiguities in the peaks of some specimens. We used the Champuru software v. 1.1 [32] (https://eeg-ebe.github.io/Champuru/input.html (accessed on 15 February 2023)) to determine the haplotypes of heterozygous individuals. While some of the substitutions, present in single specimens or haplotypes, may have been PCR artifacts, the frequent occurrence of common patterns between the specimens indicated that most of the sequence variations reflected the real ITS heterogeneity. Since the two main haplotypes of *S. lovenii* (Sl-1 and Sl-2) differed in terms of the deletion of two nucleotides (positions in

the dataset 37–38), the hybridization of these lineages resulted in hybrids with wide areas of double peaks in the ITS chromatograms. The presence of parental ITS alleles in the experimental hybrids was proven by cloning (sample H144) and by the separation of alleles using the Champuru software v. 1.1 [32] (https://eeg-ebe.github.io/Champuru/input.html (accessed on 15 February 2023)). Hybridization between the *S. lovenii* lineages in the sea is possible because the breeding period of free-swimming medusae and medusoids partially coincides [13]. However, hybridization is not the only plausible cause of the intra-individual genetic polymorphism.

Tandemly arranged gene families tend to exhibit concerted evolution, a term used to describe the phenomenon when multiple copies of a gene family tend to be homogeneous, leading to greater sequence similarities among the paralogues within a genome than among orthologues among species [52,53]. Recombinant processes such as gene conversion and unequal crossover, etc., are thought to be the homogenizing mechanisms [53–56]. Despite concerted evolution, intragenomic ITS variation has been found in many different types of invertebrates [57–62], indicating that consideration has to be given for intra-individual rDNA variation. The simplest reason for the appearance of intra-individual rDNA variation is hybridization between different species or haplotypes of the same species [43]. Significant variation between copies within a species has been also attributed to introgression from hybridization, pseudogenes, separately evolving chromosomal lineages and slowed rates of lineage sorting of ancestral alleles [51,58,63–65]. Hybridization and intragenomic rDNA polymorphism are often difficult to distinguish [66].

We suppose that the ITS polymorphism in *S. lovenii* and *S. tubulosa* is primarily associated with intraspecific hybridization. The sequence data from the ITS indicated that the rDNA arrays were homogeneous in the specimens related to the haplotypes Sl-1, Sl-2, Sl-4 and St-1, St-2. Though we did not perform the mass cloning of our DNA samples, we assume that intragenomic polymorphism was absent or insignificant for these specimens. In addition, the polymorphism of many of the specimens might be explained by the presence of hybrid forms between known haplotypes. Here, we experimentally proved that the polymorphism was a result of crossing. Vegetative reproduction is likely to be a reason for the maintenance of the parental ITS sequences in the hybrids. Thus, questions remain for those specimens with ITS polymorphism for which we did not find potential parental haplotypes. It seems unlikely to us that intragenomic polymorphism occurs in some lineages of *S. lovenii* but is absent in other lineages. However, the presence of a network of interconnected haplotypes in *S. lovenii* suggests the presence of genetic connectivity between them and the transfer of genetic material through recombinant processes.

## 5. Conclusions

Here, the morphogenetic diversity of hydrozoans *Sarsia* spp. in the White Sea was described. Four morphotypes of gonophores were identified. A new morphotype of the gonophore of *S. lovenii* (abnormal medusoid) differs from typical medusoids due to the presence of elongated rod-shaped tentacular bulbs and by an earlier period of appearance. We have shown that the morphotype attached medusa is produced by intraspecific hybrids between medusae of *S. lovenii* and medusoids of *S. lovenii* and by interspecific hybrids between medusoids of *S. lovenii* and medusae of *S. tubulosa*. We have also experimentally demonstrated the possibility of backcrossing for the interlineage hybrid F1 of *S. lovenii* and obtained descendants of F2. When analyzing the COI, we found that each haplogroup of *S. lovenii* included specimens with the morphotype attached medusa, which indicates that interlineage crossing between medusae and medusoids goes both ways. The specimen of *S. lovenii* from the Bering Sea did not fall into the existing COI haplogroups. Further research is required to understand the intraspecific diversity and phylogeography of *S. lovenii* in the Arctic seas. We also found intra-individual polymorphism in the structure of the ITS for *S. lovenii* and *S. tubulosa*. We have experimentally proven that part of the observed polymorphism of the ITS for *S. lovenii* can be explained by hybridization between frequently encountered haplotypes. In other cases, ITS polymorphism can also be explained

by crossing between different lineages. However, the contribution of other processes, such as introgression from hybridization, is not excluded. Thus, potential introgression due to hybridization, as a necessary component of reticulate evolution, is a promising direction for further research.

**Supplementary Materials:** The following supporting information can be downloaded at: https://www.mdpi.com/article/10.3390/d15050675/s1, Figure S1: Photographs of specimens used for phylogenetic analyses. Full description of specimens presented in Table S1. Scale bars 1 mm; Table S1: List of *Sarsia* specimens from the White Sea used for phylogenetic analyses: collection data and GenBank accession numbers. Abbreviations: Exp—experiment. Species: Sl—*Sarsia lovenii*, St—*Sarsia tubulosa*. Sex: Fem—female, male. Gonophore type: 1—medusoid, 2—medusa, 3—attached medusa, 4—"abnormal medusoid". Stage: p—polyp, m –medusa, pm—medusoid. Locality (see Figure 1): W1—aquarium at WSBS; W2—pier of WSBS; W3—Eremeevskie rapids; W4—saline lake at the Green Cape; W5—location "Luda"; W6—location "Krest"; W7—Rugozerskaya inlet, depth 5–15 m; W8—Polovye islands; W9—Velikaya Salma Strait, depth 40–60 m. Table S2: Specimens, haplotypes and haplogroups COI of *S. lovenii*, *S. tubulosa* and *S. princeps* visualized at haplotype net (Figure 3B). Sequences excluded from analysis of COI haplotypes are denoted in table as "exc". Morphotypes for *S. lovenii*: medusoid, medusa, attached medusa, abnormal medusoid, unknown morphotype. Table S3: Alleles of significant loci of ITS dataset associated with interspecies and haplotypes differences (see Figure 4). Table S4: Specimens and ITS haplotypes of *Sarsia lovenii* (Sl-1–Sl-14), *Sarsia tubulosa* (St-1, St-2) and *Sarsia princeps* visualized at haplotype network (Figure 4). Abbreviations: allel _1 and allel_2—haplotypes separated manually from sequences with one heterozygote. Phase1 and phase2—haplotypes separated from sequences by means of Champuru v. 1.1 (Flot, 2007). St-1-add—specimens *S. tubulosa* of haplotype 1 with additive heterozygotes in some loci. St-1/2—specimens with many single heterozygotes. Specimens of *S. tubulosa* with heterozygotes (St-1-add and St-1/2) were not included in haplotype analyses.

**Author Contributions:** Conceptualization, A.P.; methodology, A.P.; software, A.P.; validation, A.P.; formal analysis, A.P.; investigation, A.P., A.V. and S.K.; resources, A.P. and S.K.; data curation, A.P.; writing—original draft preparation, A.P.; writing—review and editing, A.P. and A.V.; visualization, A.P.; supervision, A.P.; project administration, A.P.; funding acquisition, A.P. and S.K. All authors have read and agreed to the published version of the manuscript.

**Funding:** This study was supported by the Russian Science Foundation, grant number 21-74-00129 (Stanislav Kremnyov) and the Scientific Project of the State Order of the Government of Russian Federation to Lomonosov Moscow State University, grant number 121032300118–0 (Andrey Prudkovsky). Molecular cloning was carried out as part of the implementation of grant RSF 21-74-00129. The funders had no role in study design, data collection and analysis, decision to publish or preparation of the manuscript.

**Institutional Review Board Statement:** Not applicable.

**Data Availability Statement:** All data generated or analyzed during this study are included in this published article (and Supplementary Materials) or are available from the corresponding authors upon reasonable request.

**Acknowledgments:** We thank Ivan Fedutin and Olga Filatova for collecting the specimens in the Bering Sea and we thank Nikolai Neretin, Glafira Kolbasova, Anna Mihlina and Boris Osadchenko for collecting the specimens in the Barents Sea. We would like to acknowledge the staff of N.A. Pertzov White Sea Biological Station of Lomonosov Moscow State University, Russia, for providing the opportunity for the research and the equipment usage of the Center of Microscopy WSBS. We thank the Scuba Diving WSBS team for the material collection. We are sincerely grateful to the anonymous reviewers for their critical evaluation of the manuscript.

**Conflicts of Interest:** The authors declare no conflict of interest.

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
