# Peer review of "What Does “ITS” Say about Hybridization in Lineages of Sarsia (Corynidae, Hydrozoa) from the White Sea?"

_diversity, doi:10.3390/d15050675_

Round 1
Reviewer 1 Report
The manuscript was prepared with cooperation of three scientists as invertebrate zoologist, embryologist and the specialist for the morphogenesis evolution, what is the reason for the wide scientific approach and sufficient quality of the research.
The legend for the Fig. 1 seems too expanded. Such simple schematic map doesn’t need so long references to satellites and the origin of it.
When distinguishing four morphotypes of gonophores in specimens of Sarsia lovenii from the White Sea, how (and how long time) authors controlled the further development of the each morphotype specimen? Was it really the final stage of the gonophore’s development and transformation or not yet?
Every free-swimming medusae pass the stage “attached medusa” before the liberation from a colony. The difference between the Morphotype III, "attached medusa" (Fig. 2, C) and the Morphotype II, “medusa” seems not so evident, especially if authors have mentioned that "attached medusa" “may broke away from the mother colony and swim near the bottom and even be able to feed” (p. 6, rows 214-215).
All illustrations (photographs) are of adequate quality, however the total number of them is extremely high – 137 photographs! Evidently this number can be halved without substantial loss of the total information content.
The final chapter of the article is titled “Discussion”. Some of results are really debatable. However, the chapter “Conclusion” with verified results is missing. All conclusions left for readers to make themselves and readers can do it various way. I suppose the authors should separate the hypothetical considerations from the evident conclusions in the special short final chapter of the article.
Author Response
Comment: The legend for the Fig. 1 seems too expanded. Such simple schematic map doesn’t need so long references to satellites and the origin of it.
Answer: Done
Comment: When distinguishing four morphotypes of gonophores in specimens of Sarsia lovenii from the White Sea, how (and how long time) authors controlled the further development of the each morphotype specimen? Was it really the final stage of the gonophore’s development and transformation or not yet?
Answer: We have supplemented the materials and methods, and discussion:
We observed the process of gonophores development from the moment of their appearance to the period of gonad formation or until the moment of detachment of the gonophore from the parent colony. The final stage of gonophore’s development is spawning. However, the morphology of gonophores changes little after the appearance of gonad. In some cases, we traced the development of gonophores up to spawning.
Comment: Every free-swimming medusae pass the stage “attached medusa” before the liberation from a colony. The difference between the Morphotype III, "attached medusa" (Fig. 2, C) and the Morphotype II, “medusa” seems not so evident, especially if authors have mentioned that "attached medusa" “may broke away from the mother colony and swim near the bottom and even be able to feed” (p. 6, rows 214-215).
Answer: We have supplemented the results and discussion:
The attached medusa becomes free only as a result of mechanical action after the appearance of the gonad. Most medusae broke away from the parental colony when experimental hydroids were pulled out of the aquarium for observation. The size of such medusa (morphotype III) is much smaller (1-3 mm) than the size of mature free-swimming medusa of morphotype II (7-16 mm) [13].
Comment: All illustrations (photographs) are of adequate quality, however the total number of them is extremely high – 137 photographs! Evidently this number can be halved without substantial loss of the total information content.
Answer: Indeed, there are quite a lot of photographs in the supplement. Many polyps and medusae are similar to each other. However, this information may be useful in further work when analyzing finds from other locations. Sometimes it is important to know from which specimen the DNA was extracted. Since the journal does not set limits on the number of photographs in the supplement, I consider it necessary to leave them all.
Comment: The final chapter of the article is titled “Discussion”. Some of results are really debatable. However, the chapter “Conclusion” with verified results is missing. All conclusions left for readers to make themselves and readers can do it various way. I suppose the authors should separate the hypothetical considerations from the evident conclusions in the special short final chapter of the article.
Answer: We added conclusion chapter.
Reviewer 2 Report
The manuscript is clear, relevant for the field and structured mostly appropriately. It includes present-day, adequate and sufficient amount of cited literature covering all main questions. Methods are appropriate for the objective, described in details, so the results are reproducible (if somebody would like to repeat such laborious research). Illustrations are of good quality and corroborate the text. Conclusions mostly consistent with the arguments presented. All additional but necessary data on material description are presented in Supplementary Materials.
Definitely this work will be of interest for broad audience.
The few complaints concern orthography and terms usage (mentioned below):
Line 20: “For S. lovenii, we identified …” – remove comma.
Line 46: “…the medusa and usually form gametes …” – produce gamets...
Line 79: “…manually from the surface of the water.” – near (under ) the surface...
Lines 102-104: “…if gonophore detaches from the mother polyp as a free-swimming medusa or not; if edges of the bell with tentacles are bent inward or not, if bell unfolds...” - does gonophore detaches from the mother polyp as a free-swimming medusa or not; do edges of the bell with tentacles bent inward or not, does bell unfold...
Line 122: Amplification was also used in a volume of 25 ml, – was also carried …
Lines 187 and further on within whole text: “…tentacle bulbs (Figure 2A). Tentacular bulbs are present, …” – use ‘tentacular bulbs’ as right term.
Lines 188-189: “The bell is oval, noticeably narrowed in the distal part.” – the bell is oval in shape – from what point of view? Distal part of the bell – what part is distal? And what is proximal? Apical part? Please, rewrite to clarify.
Line 189: “The manubrium is encircled by gonads” – maybe it is better “The gonad encircle manubrium”....
Line 190: “Gonads on the manubrium appear …” – Are there several gonads on the manubrium? And within whole text it is better to use this term (“gonad”) in brackets, as there there are no real organs in Hydrozoa.
Line 194: “…gametes are expelled from the subumbrellar cavity.” – But before being expelled from the subumbrellar cavity, the gametes have to get there from “gonads”.
Lines 197-198: “...detaches from the mother colony when it is completely developed.” – What does it mean ‘completely developed’?
Lines 210-211: “...and tentacle bulbs with ocellus.” – ocelli.
Line 222: “…tentacular bulbs often elongated into short rod-shaped tentacles.” – looking like short rod-shaped tentacles...
Line 327: “Specimens with medusoid were...” – medusoids
In Discussion logically it is better to put section “4.4. Hybridization experiments” before section “4.3. Hybridization or intragenomic polymorphism?”
Lines 451-452: “We thank to Ivan Fedutin and Olga Filatova for collecting specimen in Bering Sea and we thank to Nikolai Neretin...” – delete “to”.
Line 453: “...staff of N.F. Pertzov White...” - of N.A. Pertzov.
Also I would recommend to add Conclusions section at the end of the manuscript, as most of the main conclusion statements are scattered between different sections of the Discussion.
Remarks on the figures.
Figure 2, it is better to replace red arrows with magenta or white arrows.
Figure 3B – try to enlarge the legend, the letters are too minute.
English language needs minor editing (checking for misprints) - some are mentioned above.
Author Response
Comment: Line 20: “For S. lovenii, we identified …” – remove comma.
Answer: Done
Comment: Line 46: “…the medusa and usually form gametes …” – produce gamets...
Answer: Done
Comment: Line 79: “…manually from the surface of the water.” – near (under ) the surface...
Answer: Done
Comment: Lines 102-104: “…if gonophore detaches from the mother polyp as a free-swimming medusa or not; if edges of the bell with tentacles are bent inward or not, if bell unfolds...” - does gonophore detaches from the mother polyp as a free-swimming medusa or not; do edges of the bell with tentacles bent inward or not, does bell unfold...
Answer: Done
Comment: Line 122: Amplification was also used in a volume of 25 ml, – was also carried …
Answer: Done
Comment: Lines 187 and further on within whole text: “…tentacle bulbs (Figure 2A). Tentacular bulbs are present, …” – use ‘tentacular bulbs’ as right term.
Answer: Done
Comment: Lines 188-189: “The bell is oval, noticeably narrowed in the distal part.” – the bell is oval in shape – from what point of view? Distal part of the bell – what part is distal? And what is proximal? Apical part? Please, rewrite to clarify.
Answer: We used the terms proximal and distal in sense Haddock et al. (2005), in accordance with the terminology for Siphonophore. It seems to me that this terminology is quite appropriate in this case. We have supplemented the text to clarify the terminology.
Comment: Line 189: “The manubrium is encircled by gonads” – maybe it is better “The gonad encircle manubrium”....
Answer: Done
Comment: Line 190: “Gonads on the manubrium appear …” – Are there several gonads on the manubrium? And within whole text it is better to use this term (“gonad”) in brackets, as there there are no real organs in Hydrozoa.
Answer: Done. In accordance with the reviewer's comment, we put the term "gonad" in quotation marks in the introduction chapter. However, in other cases, we left the term without quotes, so as not to overload the text.
Comment: Line 194: “…gametes are expelled from the subumbrellar cavity.” – But before being expelled from the subumbrellar cavity, the gametes have to get there from “gonads”.
Answer: Done
Comment: Lines 197-198: “...detaches from the mother colony when it is completely developed.” – What does it mean ‘completely developed’?
Answer: Done. We revised the text.
Comment: Lines 210-211: “...and tentacle bulbs with ocellus.” – ocelli.
Answer: Done
Comment: Line 222: “…tentacular bulbs often elongated into short rod-shaped tentacles.” – looking like short rod-shaped tentacles...
Answer: Done
Comment: Line 327: “Specimens with medusoid were...” – medusoids
Answer: Done
Comment: In Discussion logically it is better to put section “4.4. Hybridization experiments” before section “4.3. Hybridization or intragenomic polymorphism?”
Answer: Done
Comment: Lines 451-452: “We thank to Ivan Fedutin and Olga Filatova for collecting specimen in Bering Sea and we thank to Nikolai Neretin...” – delete “to”.
Answer: Done
Comment: Line 453: “...staff of N.F. Pertzov White...” - of N.A. Pertzov.
Answer: Done
Comment: Also I would recommend to add Conclusions section at the end of the manuscript, as most of the main conclusion statements are scattered between different sections of the Discussion.
Answer: Done
Remarks on the figures.
Comment: Figure 2, it is better to replace red arrows with magenta or white arrows.
Answer: Done
Comment: Figure 3B – try to enlarge the legend, the letters are too minute.
Answer: Done
Reviewer 3 Report
The work is very interesting and overall well-done. My concerns are about the types of gonophoroes: are the authors sure that they are not different developmental stages of the same gonophore (at least the 2 medusae and the 2 medusoids)?. More clarity is needed in certain parts of the manuscript. Also I would use the term ‘hybridization’ only for phenomena occurring inter-specifically and not intra-specifically. Please find my comments below.
Introduction: Please revise English since most words lack the article
L30-31: Authors should specify that hybridization occurs between populations of different species
L42: not always shortly… it may take long time
L49: please add authorities every time you introduce a genus and species name
L58: of S. lovenii
L78: which substrates?
L95: remove ‘-‘ (here and in the rest of the manuscript)
L97-98: how did you ‘experimentally induced’ the formation of gonophores?
L100: please explain why you analyzed these features
L102: mother polyp à parental polyp
L156: remove ‘-‘
L159-167: please rephrase. It is not clear to me your strategy for phasing the sequences. Please explain better
L174: It is not clear to which species the different medusae or medusoids belong. How do you know that the two types of medusae and medusoids are not just different developmental stages or their ‘reduced’ development is due to environmental conditions? I think that this paragraph of the results needs to be rewritten giving more details.
L232: it is not clear on what basis you divide it into two haplotypes. What about the phylogenetic structure? Do they correspond to two well defined clades? If yes they should be visible in the phylogenetic tree figure. What about genetic distances?
L242-243: It is not clear, please rephrase
L257: I think it will be useful if you find a way (a table, a figure?) to summarize all the ‘hydridization’ results, otherwise they are a bit messy and difficult to follow
L322: what about Sarsia sp.? Is it belonging to one of the 3 species? If yes, there is no need to call it Sarsia sp. previously
L331: ‘which corresponds to hybrid forms’ this is not very clear from the results and should be hilghlighted in the result section in some way
L336: how can you discriminate between the influence of environmental factors and the ‘hybridization’ events in the reduction of the gonophore
minor edits required
Author Response
Comment: The work is very interesting and overall well-done. My concerns are about the types of gonophoroes: are the authors sure that they are not different developmental stages of the same gonophore (at least the 2 medusae and the 2 medusoids)?. More clarity is needed in certain parts of the manuscript. Also I would use the term ‘hybridization’ only for phenomena occurring inter-specifically and not intra-specifically. Please find my comments below.
Answer: Different morphotypes of gonophores are not different stages of development. We tracked the development process on experimental colonies. To clarify this issue we have supplemented the chapters materials and methods, results and discussion.
Regarding the term hybridization, see below.
Comment: Introduction: Please revise English since most words lack the article
Answer: We checked English
Comment: L30-31: Authors should specify that hybridization occurs between populations of different species
Answer: I don't quite agree with this remark. The term "intraspecific hybridization" is used in scientific articles: there are about 5000 references according to Google Scholar. According to the definition «Hybridization occurs when two genetically distinct individuals (that in turn can belong to different subspecies, species, genera, and even families) reproduce offspring» (Gontier, 2015). The two morphotypes of S. lovenii essentially have the rank of subspecies, due to genetic, morphological and ecological differences. Although they do not have such a taxonomic status yet. Therefore, I believe that the use of the term hybridization is justified at least for crossing between medusa and medusoid of S. lovenii. However, the definition of hybridization can be broader, according to various criteria of the species. The term “hybridization” may be used when any partial reproduction barriers appear (for example, Abbott et al., 2013). We explained our point of view in the introduction.
Comment: L42: not always shortly… it may take long time
Answer: Done
Comment: L49: please add authorities every time you introduce a genus and species name
Answer: Done
Comment: L58: of S. lovenii
Answer: Done
Comment: L78: which substrates?
Answer: The substrates are listed in the Table S1
Comment: L95: remove ‘-‘ (here and in the rest of the manuscript)
Answer: Done
Comment: L97-98: how did you ‘experimentally induced’ the formation of gonophores?
Answer: We have added a description of the methodology to the “materials and methods”
Comment: L100: please explain why you analyzed these features
Answer: We have added to the Morphological analysis: “We observed the process of gonophore development from the moment of their appearance to the period of gonad formation or until the moment of detachment of the gonophore from the parent colony. Based on the results of this observation, we selected the features for the morphotypes delimitation.”
Comment: L102: mother polyp à parental polyp
Answer: Done
Comment: L156: remove ‘-‘
Answer: Done
Comment: L159-167: please rephrase. It is not clear to me your strategy for phasing the sequences. Please explain better
Answer: We add, that Champuru is “a computer software for unraveling mixtures of two DNA sequences of unequal lengths”
Comment: L174: It is not clear to which species the different medusae or medusoids belong. How do you know that the two types of medusae and medusoids are not just different developmental stages or their ‘reduced’ development is due to environmental conditions? I think that this paragraph of the results needs to be rewritten giving more details.
Answer: We have added the species affiliation of the morphotypes. According to our results, each haplotype produces only one morphotype of the gonophore. This is confirmed statistically (for a large number of specimens) and experimentally. In the experiment, hydroids of different haplotypes produced different gonophores under the same conditions. We have already discussed this issue previously (Prudkovsky et al., 2019) and consider it unnecessary to repeat the discussion here.
Comment: L232: it is not clear on what basis you divide it into two haplotypes. What about the phylogenetic structure? Do they correspond to two well defined clades? If yes they should be visible in the phylogenetic tree figure. What about genetic distances?
Answer: These analyses were performed in a previous paper (Prudkovsky et al., 2019). The division of Sarsia lovenii into two clades was obtained for a multigenic tree. In this paper, we do not repeat this study. In addition, ITS sequences with heterozygotes, to which the reviewed work is devoted, will obviously reduce the level of support for the clades for the multigenic tree. Therefore, the network of haplotypes presented by us is more informative. However, the phylogenetic tree and p-distances required by the reviewer are presented in the article Prudkovsky et al., 2019. We have supplemented the discussion chapter to clarify this.
Comment: L242-243: It is not clear, please rephrase
Answer: We change to “The diversity of COI haplotypes of S. loveni in the White Sea is low, most of the specimens belong to the two widespread haplotypes.”
Comment: L257: I think it will be useful if you find a way (a table, a figure?) to summarize all the ‘hydridization’ results, otherwise they are a bit messy and difficult to follow
Answer: All the results are shown in Fig. 4 and in Table S4. We have put links to Table S4
Comment: L322: what about Sarsia sp.? Is it belonging to one of the 3 species? If yes, there is no need to call it Sarsia sp. previously
Answer: We change Sarsia sp. to Sarsia lovenii
Comment: L331: ‘which corresponds to hybrid forms’ this is not very clear from the results and should be hilghlighted in the result section in some way
Answer: We have changed the discussion and supplemented the results.
Comment: L336: how can you discriminate between the influence of environmental factors and the ‘hybridization’ events in the reduction of the gonophore
Answer: We have changed the discussion and supplemented the results. According to our results, each haplotype produces only one morphotype of the gonophore. This is confirmed statistically (for a large number of specimens) and experimentally. In the experiment, hydroids of different haplotypes produced different gonophores under the same conditions. We have already discussed this issue previously (Prudkovsky et al., 2019) and consider it unnecessary to repeat the discussion here.